# Is Social Training Delivered with a Head-Mounted Display Suitable for Patients with Hereditary Ataxia?

**DOI:** 10.3390/brainsci13071017

**Published:** 2023-06-30

**Authors:** Giorgia Malerba, Silvia Bellazzecca, Cosimo Urgesi, Niccolò Butti, Maria Grazia D’Angelo, Eleonora Diella, Emilia Biffi

**Affiliations:** 1Scientific Institute, IRCCS E. Medea, 23842 Bosisio Parini, Italy; 2Laboratory of Cognitive Neuroscience, Department of Languages and Literatures, Communication, Education and Society, University of Udine, 33100 Udine, Italy; 3PhD Program in Neural and Cognitive Sciences, Department of Life Sciences, University of Trieste, 34127 Trieste, Italy

**Keywords:** head-mounted display, hereditary ataxia, social training, usability, cyber sickness

## Abstract

Social cognition is fundamental in everyday life to understand “others’ behavior”, which is a key feature of social abilities. Previous studies demonstrated the efficacy of a rehabilitative intervention in semi-immersive virtual reality (VR) controlled by whole-body motion to improve the ability of patients with cerebellar disorders to predict others’ intentions (VR-SPIRIT). Patients with severe ataxia that have difficulties at multiple levels of social processing could benefit from this intervention in terms of improving their social prediction skills, but they may have difficulties in controlling VR with whole-body movements. Therefore, we implemented VR-SPIRIT on a wearable, affordable, and easy-to-use technology, such as the Oculus Quest, a head-mounted display. The aim of this work was to evaluate the usability and tolerability of this VR application. We recruited 10 patients (37.7 ± 14.8 years old, seven males) with different types of hereditary ataxia who performed a single VR-SPIRIT session using the Oculus Quest viewer. After the session, patients answered a series of questionnaires to investigate the overall usability of the system and its potential effects in terms of cyber sickness. The preliminary results demonstrated system usability and tolerability. Indeed, only three patients did not complete the session due to different problems (dizziness, nausea, and boredom). In future studies, more patients will be enrolled to assess the effectiveness of the application, paving the way for the implementation of social training that can also be delivered at home.

## 1. Introduction

Human life inherently embeds social cognition processes. People socialize daily and interact with strangers or acquaintances that can have different behaviors, beliefs, or cultures. For this reason, in everyday life it is very important to understand “others’ behavior”, which is a key feature of social abilities. Deficits in social cognition can have serious consequences on people’s relationships and activities, which are fundamental needs for happiness and wellbeing [1,2]. The capacity to interpret and predict others’ intents concerns different aspects of social cognitive processing, such as action understanding [3,4], emotion recognition [5,6], and theory of mind [7,8] abilities.

According to a predictive coding account, social cognitive processing relies on the continuous generation by the brain of predictions about incoming events, the so-called “priors”, which can arise from innate assumptions or from lived experiences [9]. Within a wide prediction network of the brain [10], the cerebellum would play a pivotal role in encoding and updating internal prediction models across various cognitive domains, from sensorimotor control to language [11,12]. While this is in line with the hypothesis of domain-general unique cerebellar computation [13,14], recent evidence suggests the specific involvement of the cerebellum in processing social sequences and in forming expectations related to social events compared to non-social, physical events [13,14,15].

Accordingly, previous studies from our group [16,17] have provided supporting evidence that cerebellar alterations impair context-based predictions of others’ intentions. Indeed, we found that, compared to patients with supratentorial alterations, patients with either congenital [16] or acquired [17] infratentorial alterations were impaired in terms of using contextual cues to predict the final outcomes of observed actions ahead of realization. Interestingly, these impairments were independent of general cognitive abilities. Furthermore, we developed and analyzed the efficacy of a rehabilitative intervention in virtual reality (VR) targeting social prediction to improve the use of contextual priors during the prediction of others’ intentions in cerebellar patients [16,17,18]. In these works, we described the design, development, and testing of VR-Social Prediction Improvement and Rehabilitation Intensive Training (VR-SPIRIT), which was specifically designed for the Gait Real-time Analysis Interactive Lab (GRAIL, Motek, Amsterdam, The Netherlands), a VR cave. In VR-SPIRIT, participants were immersed in a playground scenario and engaged in a competition with one of four avatars. Trial-by-trial, the participants learned the preference of each avatar and predicted his/her behavior. The results provided the first evidence that social prediction training in VR could improve the ability of patients with cerebellar disorders to use context-based predictions for understanding others’ intentions [16].

However, technologies like the GRAIL system have some limitations, including their size and cost, that make them available only in specialized laboratories. Furthermore, the VR-SPIRIT intervention designed for the GRAIL was ultimately unsuitable for patients with balance or severe motor impairments.

Cerebellum disorders are often associated with ataxia, a neurological disorder that affects the control and coordination of voluntary movements [19]. Beyond classical motor and oculomotor symptoms, alterations of cerebellar predictive functions result in neurocognitive impairments and behavioral–affective abnormalities described as cerebellar cognitive affective syndrome [20,21]. In keeping with the major role of prediction in social cognition [22,23], CCAS includes deficits at multiple levels of social processing, from low-level action perception [3,4] to complex theory of mind abilities [7,8]. As documented for non-progressive malformations of the cerebellum [10], individuals with hereditary ataxia could thus benefit from VR-SPIRIT. However, they would experience serious difficulties in performing the required motor tasks in the GRAIL-designed intervention.

To overcome the abovementioned limitations, we decided to implement the same training by using VR head-mounted displays (HMDs), which are wearable devices that are more affordable than VR caves and can even be used by patients with impaired motor control of the limbs. However, the literature reports some issues related to the use of this technology, among which cyber sickness is the most common effect [24].

This work aims to describe the design and development of VR-SPIRIT for HMDs and to verify the usability and tolerability of this technology in patients with hereditary ataxia who performed a single-session assessment.

## 2. Materials and Methods

### 2.1. Participants

Patients were recruited at the Scientific Institute IRCCS E. Medea. Inclusion criteria were: (i) defined diagnosis of progressive ataxia; (ii) older than 13 years old; (iii) absence of severe cognitive delay; (iv) no history of epileptic seizures; (v) head and neck control; and (vi) visual acuity that can be corrected with optical lenses. A total of 10 patients were recruited (mean age = 37.7 years; SD = 14.8 years; and 7 males). Participants provided their consent to participate in the study and signed a written informed consent, which was accompanied through the written informed consent of parents for patients aged less than 18 years old. The study procedures were in accordance with the Declaration of Helsinki and approved by the local ethical committee (IRCCS E. Medea, Protocol N. 74/21-CE, 21 September 2021).

### 2.2. Hardware Solution

HMDs are headsets used to display VR, with several applications. There are several types of HMDs: we chose the Oculus Quest viewer on the basis of a literature and market analysis.

The Oculus Quest is a portable HMD with two controllers equipped with motion sensors that are used to track hand position. It is a standalone device that can run software wirelessly by using an Android-based operating system. It supports positional tracking with six degrees of freedom, using internal sensors and an array of cameras in the front of the headset. A diamond Pentile OLED display is used for each eye, with an individual resolution of 1440 × 1600 and a refresh rate of 72 Hz. It is possible to connect the viewer to a PC via a USB port to allow data storage. The Oculus platform gives users the opportunity to develop their own application programs on a software platform like Unity 3D (Unity Technologies, San Francisco, CA, USA).

### 2.3. Unity 3D Environmental Setup

Unity 3D is a software, developed by Unity Technologies, that can be obtained directly from the official website [25] through an installation process on an adequate PC with separate graphics card specifications and at least 8/16 GB RAM to support 3D graphics processing. It can be used to create three-dimensional (3D) and two-dimensional (2D) games, as well as interactive simulations and other experiences.

We used Unity 3D to realize the scenes and the applications. In particular, we realized a first application to help familiarize a user with the viewer and motion sensors, and a second application to reproduce a VR-SPIRIT session.

The first scenario was designed to instruct a user to interact with a virtual environment. The participant was asked to walk along a street in a virtual farm, using the joystick wheel to move forward and rotating their head to turn left and right. In addition, some closed gates were placed along the street to educate the user to interact with virtual objects.

The second scenario represents a VR-SPIRIT session. The setting reproduces a playground. The playground presents a linear 9 m-long street splitting into three 3 m-long streets, with three games (a slide, a rocking caterpillar, and a spinner) located in a semicircle at the same distance from the starting point. We added more details in the scene, such as trees and paths, with respect to the scenario used for the GRAIL. As in the VR-SPIRIT designed for the GRAIL system, the user has to compete with one of four avatars, two females and two males. The avatars were imported from the Unity Store and were animated with a realistic gait motion.

### 2.4. Software

The application algorithms were developed using the script module, which allows the implementation of new algorithms in the C# language.

Firstly, we implemented a script module to reproduce the same characteristics of the GRAIL-designed VR-SPIRIT session. Each avatar has a pre-established probability to move toward one of the three described games. In particular, three avatars head towards their preferred object in 80% of trials, while they choose each of the other two objects in 10% of trials. The fourth avatar moves in a random modality because object associations are set at 33%. Each session is composed of 80 trials. For each trial, avatars and their probabilistic associations are saved in a .csv file. The script module reads the .csv file to run the session chosen by the operator.

In each trial, the participant competes with one avatar at a time and s/he is asked to reach the game before the avatar. At the beginning of each trial, a countdown starts and the user can see and recognize the challenger avatar on his/her left. The avatars do not provide evidence of their intentions (i.e., directing toward one of the objects) until the crossroad. Furthermore, after the crossroad, the participant cannot pass the avatar anymore because equal maximum speeds are set. In this way, participants are forced to anticipate the avatar’s movements and, thus, to learn the probabilistic associations between the avatar and the most chosen game. The user can look around by moving his/her head to see the object reached by the avatar, also learning the avatar’s preferences in unsuccessful trials. Indeed, at the end of each trial, the subject has six seconds left to see the avatar’s chosen game; then the trial is interrupted and a new trial is presented.

A second script module controls custom actions and interactions of the user with some objects in the environment. In particular, the user has to touch a white cube in front of him/her to run each trial and s/he can defeat the avatar by reaching the game preferred by the avatar before it and animating the game with the movement of the joystick (e.g., a ball rolling over the slide).

A last module creates, trial-by-trial, a .csv file. Games chosen by the user and the avatar, as well as the instance of their arrival and the duration of the trial, are stored.

### 2.5. Experimental Procedure

#### 2.5.1. Patient Baseline Assessment

After recruitment, each participant underwent a baseline assessment during which we collected the following data: sex, age at evaluation, age at disease onset, diagnosis, functional capability evaluated through the Scale for the Assessment and Rating of Ataxia (SARA) [26], and evaluation of the potential presence of visual problems, such as nystagmus or retinopathy.

#### 2.5.2. Familiarization with the Virtual Environment

Before the VR session, after a short explanation, patients were instructed to navigate the virtual environment with the first scenario. This phase lasted about 10 min, depending on a patient’s performance.

#### 2.5.3. VR-SPIRIT Session

After the initial period of familiarization with the technology, each participant was asked to perform a single VR-SPIRIT session using the Oculus Quest viewer. Each session lasted approximately half an hour. Patients sat comfortably and they could decide to stop the session or pause at any time. The researchers collected their potential comments. Different indices about a patient’s performance were recorded automatically by the system during the session, such as the duration of each trial and the prediction score (percentage of trials in which the participant correctly predicted an avatar’s intention).

#### 2.5.4. Questionnaire Administration

After the VR-SPIRIT session, patients answered a series of questionnaires to investigate system usability and tolerability, sense of presence, and emotional wellbeing. The following questionnaires were administrated:SUS—System Usability Scale [27]. The primary outcome of the study, which evaluates the usability of the system. The SUS is a simple, ten-item five-point Likert scale giving a global view of subjective assessments of usability.SEQ—Suitability Evaluation Questionnaire [28]. The questionnaire is especially designed to test VR systems, and it evaluates both sense of presence and tolerability to detect problems frequently associated with virtual rehabilitation systems. The SEQ includes 14 questions, 13 of them with a response graded on a 5-point Likert Scale, and a last open question asking if the subject felt uncomfortable alongside the reasons for this. Questions one to seven can be clustered to assess “system evaluation”: they measure enjoyment, sense of being in the system, feeling of success and control, realism, easy-to-understand instructions, and general discomfort, as explained in [28]. Questions eight to ten are related to cyber sickness issues. Finally, the last two questions are focused on the difficulty, respectively, of the task and of the physical interface used in the system.SSQ—Simulator Sickness Questionnaire [29]. The questionnaire evaluates the effects of cyber sickness. Participants have to score 16 symptoms on a four-point scale (0–3). The symptoms can be placed into three general categories: oculomotor, disorientation, and nausea. Weights are assigned to each of the categories and summed together to obtain a single score.ITC-SOPI-Independent Television Commission-Sense of Presence Inventory [30]. It is a 5-point Likert scale, composed of 44 items. This questionnaire evaluates the user experience to understand if the person enjoyed the game, the setting, and the overall experience. In particular, the questionnaire addresses four factors: spatial presence, engagement, realness of the environment, and side effects (cyber sickness).Short-form of PANASs—Positive Affect and Negative Affect Scales [31]. In the questionnaire, the participant must evaluate how much s/he feels in the way described by the adjective when filling out the questionnaire, responding on a 5-point Likert scale. The PANAS provides measures of Positive Affects (PAs) and Negative Affects (NAs). The PA and NA scores are the sums of the ratings of the PA items and the NA items, respectively. PA indicates the level of pleasurable engagement with the product, while NA is a general factor of subjective distress.

### 2.6. Statistical Analysis

All of the statistical analyses were performed in MATLAB (R2022b, The MathWorks, Inc., Natick, MA, USA). Kendall’s or Spearman’s coefficient was performed to assess correlation among the following:Data collected during the baseline assessment and the questionnaires investigating system usability and tolerability, sense of presence, and emotional wellbeing.Baseline assessment and indices about a patient’s performance automatically collected during the trial.Questionnaires and indices of a patient’s performance.All of the questionnaires.

In particular, Kendall’s τ was used in cases of nominal variables, while Spearman’s rho was used in cases of discrete variables. The significance level was established at *p* = 0.05. Only significant correlations are reported in the Section 3.

## 3. Results

### 3.1. VR-SPIRIT Application Testing

Patients appreciated the graphic of the application. In contrast to the VR-SPIRIT application designed for the GRAIL system, the environment was visually more interesting and rich in detail. Figure 1A shows the environment designed to instruct the participants to interact with the system. Figure 1B shows VR-SPIRIT environment, with the three streets after the crossroad, ending with three games: a slide, a rocking caterpillar, and a spinner.

Trees, hedges, and paths were very realistic. The custom actions and interactions of the user with objects in the environment were adequately implemented. The user could animate the chosen game by moving the joystick and each game had a different animation: a ball rolled over the slide, the rocking caterpillar swung back and forth, and the spinner turned. All of the patients managed to interact with the virtual objects and to move in the environment via using the joystick and rotating their head. The viewer was not regarded as too heavy and all of the patients were able to easily wear it. Figure 2 shows one patient during a VR-SPIRIT session.

### 3.2. Subjects’ Performance

Table 1 shows the characteristics of each patient. The group was composed of seven males and three females, with ages ranging between 18 and 58 years old. The mean value of the SARA score was 19.6 with a standard deviation of 3.7, meaning that the group was moderately compromised. Seven patients had nystagmus and were short-sighted; one patient (S6) had slight double vision and slight nystagmus; one patient (S8) had nystagmus and a precarious visual fixation; and only one patient (S7) had no visual impairment.

Three patients did not conclude the training session due to different reasons. S6 asked to stop the test after 17 trials due to dizziness, while S7 asked to stop the session after 16 trials due to nausea and a general sense of being unwell. Conversely, patient S9 did not finish the session because of boredom, and he completed only 58 trials out of the 80 of the whole evaluation. Table 2 reports the results of the VR-SPIRIT session with the Oculus Quest for the seven patients that completed the training session. For each patient, we considered the mean duration of each trial, which could last a maximum of 21 s, and the prediction score, which was computed as the percentage of trials in which the participant correctly predicted an avatar’s intention.

### 3.3. Usability and Tolerability

Questionnaires were administered to all of the patients, regardless of whether they concluded the training session. Results are reported in Figure 3, Figure 4, Figure 5, Figure 6 and Figure 7.

Figure 3 shows the SUS results. The SUS score was computed following authors’ instructions [27], and it ranged from 0% to 100%. In particular, according to [32], the SUS was “best imaginable” for S2, S3, and S8, “excellent” for S1, S5, and S6 (who did not finish the session because of dizziness), “good” for S4, S10, and S7 (who did not finish the session because of nausea), and “ok” for S9 (who did not finish the session because of boredom). The median value of the SUS was 72.5, which corresponds to “excellent” in the adjective rating scale [32].

Figure 4 represents boxplots related to SEQ results. Scores related to each question cluster range from 0% to 100%, and the color of the scatters refers to different patients, as explained for Figure 3. The median value of the system evaluation was 79.2%; S9 gave the lowest score. The VR issue score had a median value of 6.3%, while the task and system difficulty median scores were 0%. As regards the open question of whether patients felt uncomfortable, only S6 and S7 answered “yes”. S6 indicated disorientation issues and loss of coordination, while S7 defined the environment as “too realistic”.

Results of the SSQ are reported in Figure 5. Three distinct symptom clusters (oculomotor, disorientation, and nausea) were computed, following the instructions given by the authors [29]; each symptom has a score that ranged from 0% to 100%. The SSQ revealed that S1, S2, and S8 had no symptoms of cyber sickness; S6 and S7 had quite intense cyber sickness with disorientation and nausea, respectively, as the main problem. All of the other patients had mild symptoms mostly related to oculomotor disturbances.

The results of ITC-SOPI are reported in Figure 6. Spatial presence, engagement, realness, and side effects are the four different factor scores computed calculating a mean of all completed items contributing to each factor. Each factor score ranges from 1 to 5. The median values of the different factors were, respectively, 3.35, 3.35, 3.4, and 1.17. The color of the scatters refers to different patients, as explained for Figure 3.

Results of the short-form PANASs are reported in Figure 7. The boxplot on the left is related to items of the PA scale, while the boxplot on the right represents the NA. The PA and NA score ranged from 5 to 25. The median values of the PA and NA scales were, respectively, 18.5 and 6.5.

### 3.4. Correlations Assessment

#### 3.4.1. Correlation among Baseline Assessments and Questionnaires

The Kendall’s τ correlation analysis showed a statistically significant moderate correlation (τ = 0.58) between patients’ disease and the SUS score. The SUS score of patients with Friedreich’s ataxia was overall higher than that of patients with spinocerebellar ataxia. The other correlations did not yield statistically significant results.

#### 3.4.2. Correlation among Baseline Assessments and Indices of Patients’ Performance

Baseline assessments did not correlate with the mean duration of the trial nor the prediction score.

#### 3.4.3. Correlation among Questionnaires and Indices of Performance

A statistically significant correlation emerged between the system evaluation item of the SEQ questionnaire in addition to both the mean duration (Spearman’s rho = −0.67) and the prediction score (Spearman’s rho = 0.64). The mean duration index also showed a moderate correlation (Spearman’s rho = 0.64) with the Nausea symptom of the SSQ, while the prediction score inversely correlated (r = −0.65) with the NA scale of the PANASs. The other correlations did not yield statistically significant results.

#### 3.4.4. Correlation among Questionnaires

The Spearman analysis showed a series of statistically significant correlations among the questionnaires.

The SUS score strongly correlated with the system evaluation item of the SEQ (rho = 0.86) and with the engagement item of the ITC-SOPI (rho = 0.87). Furthermore, it showed a moderate negative correlation with the nausea symptom of the SSQ (rho = −0.67).

The system evaluation item of the SEQ also showed a significant correlation with the engagement item of ITC-SOPI (rho = 0.73). Furthermore, it showed a moderate negative correlation with the nausea symptom of SSQ (rho = −0.67). The VR issue construct of the SEQ showed a correlation with the spatial presence (rho = 0.76), realness (rho = 0.92), and side effects (rho = 0.71) of ITC-SOPI, the oculomotor disturbances (rho = 0.79) and disorientation (rho = 0.72) symptoms of the SSQ, and a moderate correlation (rho = 0.67) with the NA scale of the PANASs. The task difficulty assessment of the SEQ showed a moderate correlation (rho = 0.64) with the disorientation symptom of the SSQ. The latter was significantly correlated with the NA scale of the PANASs (rho = 0.65), while the oculomotor disturbances of the SSQ were significantly correlated with the realness (rho = 0.67) and side effects (rho = 0.74) of ITC-SOPI, as well as with the NA scale of the PANASs (rho = 0.64)

The heat map below summarizes the within-questionnaire correlation results (Figure 8).

## 4. Discussion

The purpose of this work was to describe the design and development of social training realized for HMDs. Furthermore, we wanted to verify preliminarily the usability and tolerability of this technology in patients with hereditary ataxia. To the best of our knowledge, our application is the first intervention targeting social skills designed for HMDs.

We used the Oculus Quest HMD and Unity 3D software to design avatars and a playground scenario. We realized a more realistic environment and natural gait movement of avatars compared to the graphics of the previous GRAIL application.

The usability and tolerability of the system were tested in ten patients with hereditary ataxia who participated in the experiment. Three of these patients did not complete the entire session due to different problems. In particular, two of them reported cyber sickness issues and one asked to stop the session because of boredom.

Nine patients evaluated the system as intuitive and easy to use, as evidenced by questionnaire results. Even the two patients who reported cyber sickness evaluated the system as “good” and “excellent” in the SUS, and with relatively high levels of realness, spatial presence, and engagement in the ITC-SOPI questionnaire. The patient who did not conclude the entire session due to boredom did not appreciate the system overall, as highlighted by the questionnaire results. On the other hand, he did not consider it difficult to either use the system or perform the task; furthermore, he did not have any cyber sickness problems. Instead, VR issues for S6 and S7 were well underlined by the ITC-SOPI, SEQ, and SSQ results. Previous studies indeed reported cyber sickness among the main issues related to the use of headsets [24,33]. The correlation analysis highlighted that cyber sickness effects were correlated with the mean duration of the test (i.e., longer trial, greater degree of symptoms), but also with sense of presence and realness. These findings may be considered in the design of future applications to reduce cyber sickness. In general, system evaluation, as measured by the SEQ, was high, and it was correlated with shorter trials and higher performance.

Previous [34,35,36,37,38] works evaluated a rehabilitative VR task for different populations (e.g., healthy adults and patients with stroke, multiple sclerosis, or cerebral palsy). In most of these studies, system usability results were overall significantly higher than those in our work [34,35,36,37,38]. This is likely due to differences in the type of tasks used by these applications compared to ours. Indeed, refs. [34,35,36] used virtual reality to simply create a more stimulating environment while performing a motor task, while applications developed by [37,38] consisted of a repetitive motor or cognitive task. In our application, patients had to compete with four avatars and learn their preferences towards one of the pieces of recreational equipment. The usability of our application may have been influenced by the difficulty of this task. A study more comparable to ours is [39]. The session indeed consisted of 60 training trials (with a break every 15 consecutive trials) and 5 evaluation trials in which the authors used visual error augmentation to exaggerate the participants’ forward-reaching asymmetry between the end location of the hands during experimental trials, thus challenging the participants. They enrolled 12 healthy participants and 5 patients with cerebral palsy (CP). Compared to our results, the median value of the SUS scores was comparable for healthy participants (71.25) and lower for subjects with CP (57.5). System usability was correlated with the engagement of the participants, as measured by ITC-SOPI. Therefore, the implementation of different strategies to maximize engagement may be useful in improving system usability.

As regards the tolerability of the system, we did not find any study where participants did not conclude the entire session due to cyber sickness symptoms. However, due to ataxia and motor impairments, in our study participants were still and seated during the session while they seemed to move in the virtual environment. In contrast, in the other studies participants either moved using a treadmill [34,35] or were stationary [36,37,38,39] in both the virtual and real environments. Furthermore, sessions in the analyzed studies were shorter and this may have reduced cyber sickness problems, as also suggested by our results. Finally, short-form PANASs had a median value of 18.5 and 6.5 for the PA and NA scales, respectively. Low PA scores reflect “sadness and lethargy’”, whereas high PA scores reflect “high energy, full concentration, and pleasurable engagement” [31]. In a similar vein, low NA scores describe “a state of calmness and serenity”, whereas high NA scores suggest ‘subjective distress and unpleasable engagement’ [31]. It is interesting to note that the patient who reported getting bored showed the lowest achievable score for both the PA and NA scales, in keeping with a previous study showing that this happens in boring situations [40]. On the other hand, the two patients who exhibited cyber sickness symptoms had the highest score in the NA scale. Different studies indeed reported a correlation between PANAS results and cyber sickness symptoms [41,42], as also highlighted by the correlation analysis for the NA scale, which was correlated with the VR issue item of the SEQ as well as the disorientation and oculomotor disturbances items of the SSQ. Interestingly, the NA scale also inversely correlated with the prediction score indices, indicating that lower engagement was correlated with lower performance.

The correlation analysis evidenced some key features of the proposed application. Firstly, the application seems to be usable by patients of different ages and motor phenotypes, as was demonstrated by the absence of a correlation between the SARA score and the usability as well as tolerability questionnaires. Furthermore, the SARA did not show any correlation with indices of a patient’s performance, highlighting that the ability to perform a social task is not influenced by the motor abilities of a patient. Interestingly, the group of patients enrolled in this study spanned from patients with autonomous walking to patients that have used wheelchairs for a long time, demonstrating a potential impact in clinics.

Secondly, the system usability results were higher for subjects with Friedreich’s ataxia with respect to subjects with spinocerebellar ataxia. Even if these are preliminary results and the limited sample size prevents generalization, a difference in usability could be related to the different cognitive abilities that the two groups had. However, these were only assessed qualitatively through clinical experience. In the future, a standardized cognitive assessment may be useful to further investigate this aspect and adapt the application characteristics to different subjects. The customization of the application is also important because the patients’ performance was strictly related to system usability, as shown by the correlation analyses between the system evaluation item of the SEQ and both indices of performance.

Future studies will enroll more patients to better deepen the tolerability of this application, also involving healthy participants not only to have a complete evaluation of the system but also to collect normative data. New strategies will be implemented to alleviate cyber sickness symptoms. Nausea symptoms were indeed inversely correlated with the system evaluation item of the SEQ and SUS. Therefore, reducing cyber sickness effects would provide a better user experience and system evaluation. Alternating, for instance, the VR spirit training with a more relaxing and static VR scenario or with an activity without a viewer could alleviate VR issues or make the session less boring. Undesirable effects may be limited, allowing the participant to get used to VR, organizing initial sessions in which the participant interacts with simpler VR scenarios. VR issues could be related to the spatial presence and realness of the application, as shown by the correlation results. It may be useful, therefore, to implement less realistic movements and interactions with the environment.

In the future, a randomized controlled trial could be performed to evaluate the efficacy of the VR-SPIRIT provided by HMDs. Preliminary results shown in this work are encouraging for the future perspectives of this application, which has the potential of being a tool for the home training of social interaction abilities.

## 5. Conclusions

The present study provided the design and development of social training realized for HMDs. We implemented VR-SPIRIT well on a wearable, affordable, and easy-to-use technology: the Oculus Quest, a head-mounted display. We preliminarily demonstrated the usability and tolerability of this application. The questionnaire results evidenced an overall good usability of the system, including for patients with severe motor disabilities; patients did not report problems with the use of the joystick or the viewer, and the interaction with the environment was easy. Of the subjects, 90% appreciated the environment and the application; only one patient became bored after 58 trials. The tolerability results were acceptable; indeed, only 20% of the subjects experienced cyber sickness symptoms that could be related to the high realism of the application, which also provided high sense of presence, and to the duration of the session. Future works will advance the assessment of the effectiveness of this application, it being the case that is the first application for social training implemented for HDMs, thus allowing the administration of VR-SPIRIT intervention everywhere, even at home.

## Figures and Tables

**Figure 1 brainsci-13-01017-f001:**
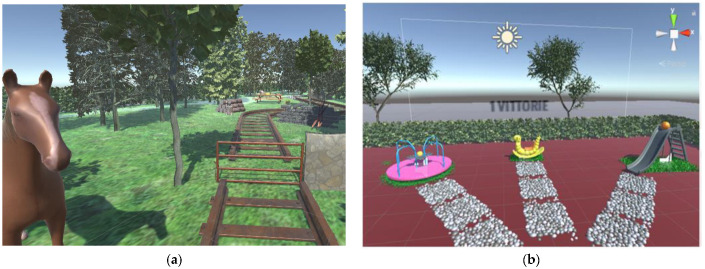
Virtual environment in Unity 3D. (**a**) First scenario representing a farm, designed to instruct user to interact with the virtual environment. (**b**) Second scenario reproducing a playground; it represents one VR-SPIRIT session.

**Figure 2 brainsci-13-01017-f002:**
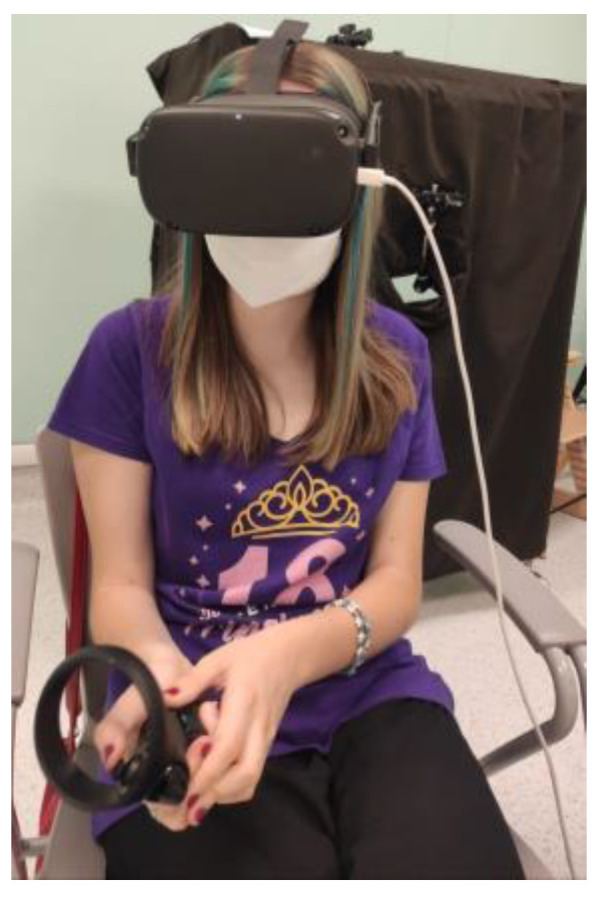
A patient undergoing a VR-SPIRIT session.

**Figure 3 brainsci-13-01017-f003:**
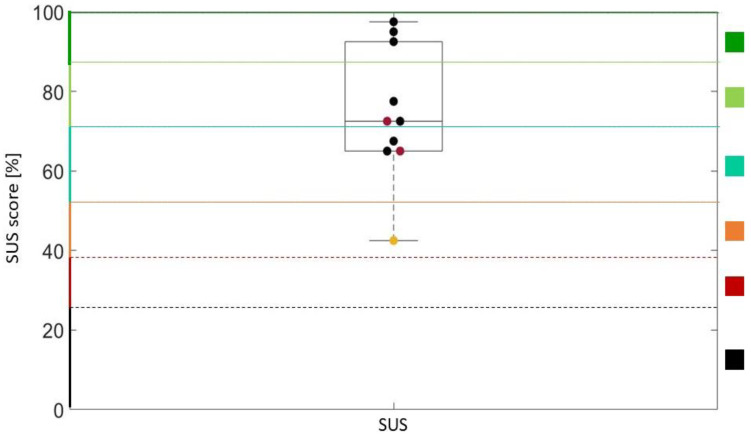
SUS results. Black scatters represent patients who completed the entire session, red scatters represent S6 and S7, who were not able to conclude all the trials because of cyber sickness symptoms, and the yellow scatter represents S9, who did not finish the session because of boredom. SUS score is divided into “best imaginable” (dark green), “excellent” (light green), “good” (green water), “ok” (orange), “poor” (red), and “worst imaginable” (black) according to [32].

**Figure 4 brainsci-13-01017-f004:**
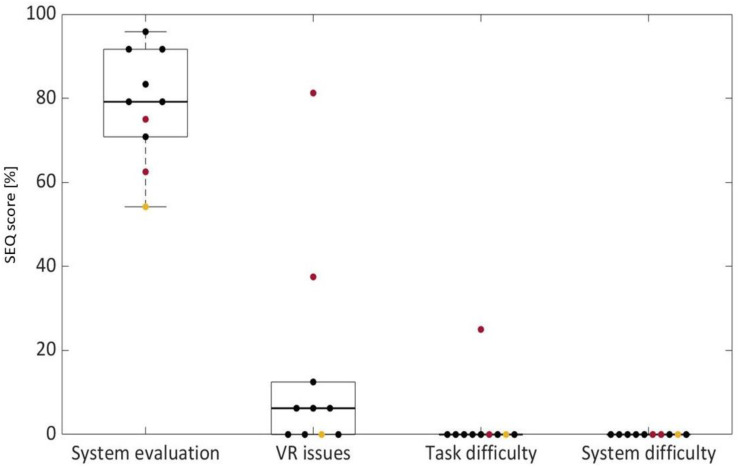
SEQ results. Black scatters represent patients who completed the entire session, red scatters represent S6 and S7, who were not able to conclude all of the trials because of cyber sickness symptoms, and the yellow scatter represents S9, who did not finish the session because of boredom.

**Figure 5 brainsci-13-01017-f005:**
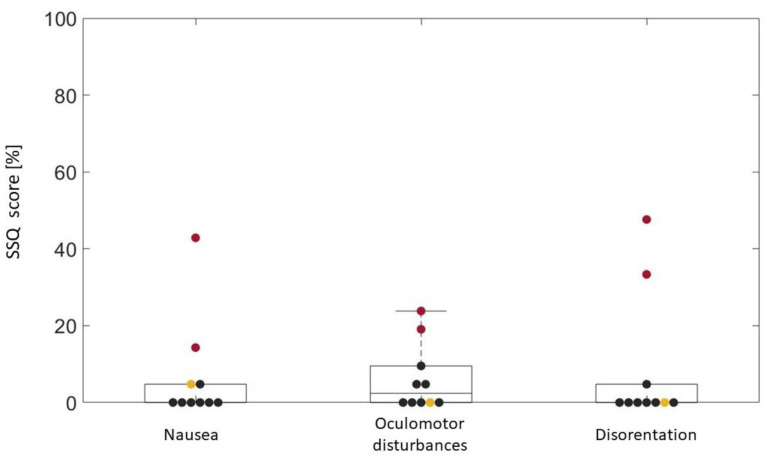
SSQ results. Black scatters represent patients who had completed the entire session, red scatters represent S6 and S7, who were not able to conclude all of the trials because of cyber sickness symptoms, and the yellow scatter represents S9, who did not finish the session because of boredom.

**Figure 6 brainsci-13-01017-f006:**
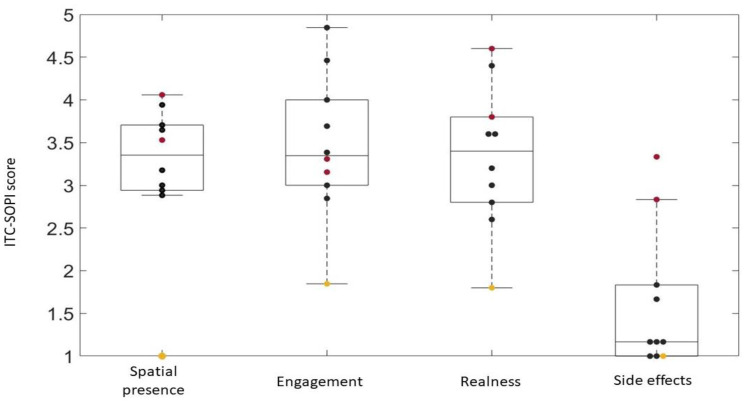
ITC-SOPI results. Black scatters represent patients who completed the entire session, red scatters represent S6 and S7, who were not able to conclude all of the trials because of cyber sickness symptoms, and the yellow scatter represents S9, who did not finish the session because of boredom.

**Figure 7 brainsci-13-01017-f007:**
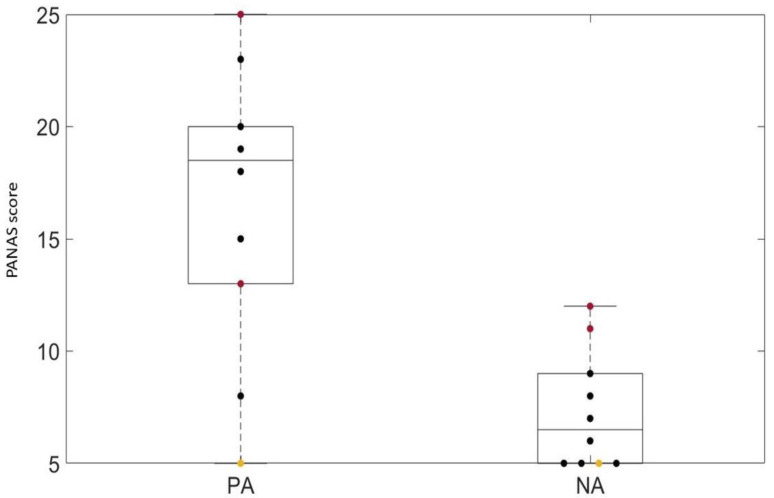
Short-form of PANAS results. The boxplots represent the Positive Affect scale (**left**) and Negative Affect scale (**right**). Black scatters represent patients who completed the entire session, red scatters represent S6 and S7, who were not able to conclude all of the trials because of cyber sickness symptoms, and the yellow scatter represents S9, who did not finish the session because of boredom.

**Figure 8 brainsci-13-01017-f008:**
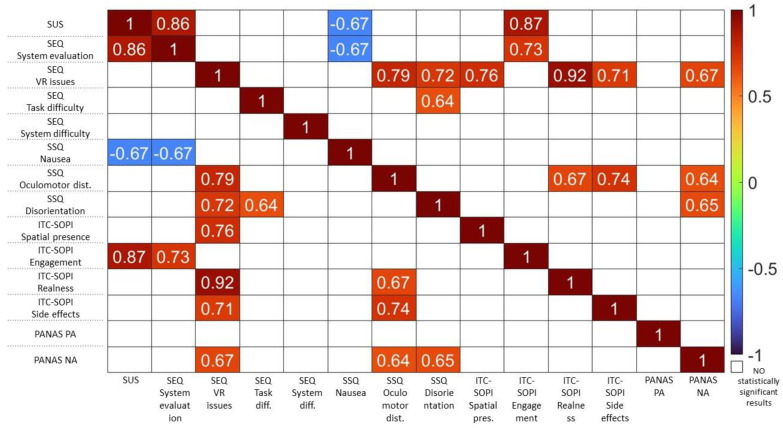
Heat map to summarize within-questionnaire correlation results. The color bar indicates the Spearman’s r coefficient range. White cells show correlations that did not perform statistically significant results.

**Table 1 brainsci-13-01017-t001:** Patients’ characteristics.

Patient	Age (Years)	Gender	SARA Score	Disease	Disease Onset (Years Old)
S1	21.0	F	17.5	Friedreich’s ataxia	9
S2	18.8	M	13.5	Friedreich’s ataxia	11
S3	28.7	M	23.5	Ataxia of Charlevoix–Saguenay	3
S4	49.9	M	22	Spinocerebellar ataxia	23
S5	51.6	M	18	Spinocerebellar ataxia	21
S6	31.4	M	19	Friedreich’s ataxia	16
S7	46.9	M	19.5	Spinocerebellar ataxia	40
S8	58.7	F	27	Friedreich’s ataxia	39
S9	51.9	M	20	Spinocerebellar ataxia	44
S10	18.3	F	15.5	Spinocerebellar ataxia	1

**Table 2 brainsci-13-01017-t002:** Patients’ performance.

Patient	Duration (s)	Prediction Score (%)
S1	16.7 (0.8)	41.3 (18.5)
S2	15.6 (0.7)	95.0 (8.7)
S3	17.2 (1.1)	61.7 (7.6)
S4	17.0 (1.0)	66.7 (30.1)
S5	11.1 (2.1)	61.7 (29.3)
S8	11.8 (2.4)	55.0 (13.2)
S10	11.9 (2.1)	71.7 (2.9)

## Data Availability

Data are available in Zenodo.

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
