# Peer review of "Is Social Training Delivered with a Head-Mounted Display Suitable for Patients with Hereditary Ataxia?"

_brainsci, 2023, doi:10.3390/brainsci13071017_

Round 1

Reviewer 1 Report

The paper provided the design and the development of a social training realized for HMD, as well as verify its usability and tolerability in HA patients. This is meaningful for the home-training of social-interaction abilities for HA patients. Several points need to be further improved.

1. May the authors provide the comparative results of the effects before and after training?

2.The statistical methods should be supplemented.

Reviewer 2 Report

The current manuscript assess the effects of social training delivered with a head mounted display in patients with hereditary ataxia. Please find the following comments: 

1- The used form is not for Brain Sciences Journal 

2- The abstract is not sound, and very hard to understand. It is need to be fully revised. 

3- I did not agree with the following '' Humans are, by nature, social animals''

4- Lines 63-65: ''To the best of our knowledge, no study besides [18,20] has designed and tested an intervention with these aims''. Revise

5- The introduction needs to be revised in shorter and easy-reading form. 

6- Why did you select the SARA in primary assessment? SARA mainly assess motor performance through different ICF domains {Etoom, M.et al Ataxia Rating Scales: Content Analysis by Linking to the International Classification of Functioning, Disability and Health. Healthcare 202210, 2459} 

7- Discussion: There is no clear explainations for the results. I did not find any clinical or research recommendations 

8- Revise the conclusion

Reviewer 3 Report

Having read the manuscript, I think the authors have done an acceptable job. However, there are still a few minor corrections to be made to ensure that the work is of the highest quality.

- The authors absolutely must find a title for the ordinate axis of figures 3, 4, 5, 6, and 7.

- Finally, in the discussion, the authors need to reflect on more elaborate perspectives.  For example, the subjects of their study are people with hereditary ataxia. The question arises as to whether people without pathologies will have the same results. I wonder to what extent the authors can improve their tool to avoid undesirable effects. All these questions should be serious avenues of research in the discussion. I therefore suggest that the authors include a paragraph on the new perspectives of their work.

Round 2

Reviewer 2 Report

Dear authors 

Thank you for addressing the comments. My only note is the abstract. it discusses too much the limitations in previous VR interventions 
